# Probing Polarity and pH Sensitivity of Carbon Dots in *Escherichia coli* through Time-Resolved Fluorescence Analyses

**DOI:** 10.3390/nano13142068

**Published:** 2023-07-14

**Authors:** Gilad Yahav, Shweta Pawar, Anat Lipovsky, Akanksha Gupta, Aharon Gedanken, Hamootal Duadi, Dror Fixler

**Affiliations:** 1Institute of Nanotechnology and Advanced Materials, Faculty of Engineering, Bar-Ilan University, Ramat Gan 52900, Israel; 2Institute of Nanotechnology and Advanced Materials, Department of Chemistry, Bar-Ilan University, Ramat Gan 52900, Israel; km-akanksha.gupta@biu.ac.il (A.G.); gedanken@mail.biu.ac.il (A.G.)

**Keywords:** carbon dots (CDs), fluorescence lifetime imaging microscopy (FLIM), time-resolved fluorescence anisotropy imaging (TR-FAIM), frequency domain (FD), *E. coli*, pH sensor, polarity sensor, intracellular sensing

## Abstract

Intracellular monitoring of pH and polarity is crucial for understanding cellular processes and functions. This study employed pH- and polarity-sensitive nanomaterials such as carbon dots (CDs) for the intracellular sensing of pH, polarity, and viscosity using integrated time-resolved fluorescence anisotropy (FA) imaging (TR-FAIM) and fluorescence lifetime (FLT) imaging microscopy (FLIM), thereby enabling comprehensive characterization. The functional groups on the surface of CDs exhibit sensitivity to changes in the microenvironment, leading to variations in fluorescence intensity (FI) and FLT according to pH and polarity. The FLT of CDs in aqueous solution changed gradually from 6.38 ± 0.05 ns to 8.03 ± 0.21 ns within a pH range of 2–8. Interestingly, a complex relationship of FI and FLT was observed during measurements of CDs with decreasing polarity. However, the FA and rotational correlation time (*θ*) increased from 0.062 ± 0.019 to 0.112 ± 0.023 and from 0.49 ± 0.03 ns to 2.01 ± 0.27 ns, respectively. This increase in FA and *θ* was attributed to the higher viscosity accompanying the decrease in polarity. Furthermore, CDs were found to bind to three locations in *Escherichia coli*: the cell wall, inner membrane, and cytoplasm, enabling intracellular characterization using FI and FA decay imaging. FLT provided insights into cytoplasmic pH (7.67 ± 0.48), which agreed with previous works, as well as the decrease in polarity in the cell wall and inner membrane. The CD aggregation was suspected in certain areas based on FA, and the *θ* provided information on cytoplasmic heterogeneity due to the aggregation and/or interactions with biomolecules. The combined TR-FAIM/FLIM system allowed for simultaneous monitoring of pH and polarity changes through FLIM and viscosity variations through TR-FAIM.

## 1. Introduction

In recent years [1,2], there has been significant interest in measuring intracellular pH and polarity due to their crucial roles in physiological and pathological processes. pH affects cell growth, apoptosis, ion transport, and homeostasis, while polarity influences cell signaling, membrane transport, enzymatic reactions, and protein interactions. Understanding cellular polarity is particularly important for drug delivery and efficacy.

Carbon dots (CDs) are a family of fluorescent carbon-based nanostructures that exhibit unique optical properties [3]. Their composition, influenced by precursors and reaction conditions, typically includes carbon, oxygen, hydrogen, and nitrogen [4]. The fluorescence emission of CDs arises from conjugated domains [5], surface defects and states [6], and molecular states [7]. By modifying the synthesis process or raw materials, one can control the luminous properties of CDs, including fluorescence intensity (FI) and emission wavelength [8]. Fluorescent probes, such as organic dyes [9], CDs [10], and semiconductor quantum dots [11], are widely used for pH and polarity [2] detection in vitro and in living systems due to their simplicity, selectivity, sensitivity, real-time monitoring capabilities, and high temporal–spatial resolution. Although organic fluorescent dye-based pH sensors have demonstrated high efficiency, they face challenges like complex synthesis and label processes, photo-bleaching, and potential cell damage [12]. In contrast, CDs offer distinct advantages as fluorescent probes. They exhibit exceptional photostability, biocompatibility, low toxicity, chemical inertness, stable photoluminescence, reduced cellular interference, and tunable fluorescence properties [13]. The distinctive characteristics of CDs make them highly promising for accurately monitoring changes in pH and polarity within biological environments. These capabilities extend to critical applications such as bacterial detection and characterization, which are essential in healthcare, food, and environmental industries, among others [14,15]. Today, various methods are employed for bacterial identification, including culture-based, immunological, and molecular techniques [16]. Culture-based methods are time-consuming [17], while immunological approaches, like enzyme-linked immunosorbent assay (ELISA), offer faster results but with lower sensitivity [18]. Molecular methods such as polymerase chain reaction (PCR) provide high sensitivity, but require specialized equipment [19].

In the field of bacterial identification, *Escherichia coli* (*E. coli*) is of particular interest. *E. coli*, a Gram-negative bacterium, is a versatile rod-shaped bacterium, ranging from 1–3 μm in length and 0.25–1.0 μm in diameter [20]. It fulfills both beneficial and pathogenic roles, naturally inhabiting the human gut, while specific strains can cause severe infections [21]. Due to its rapid growth, ease of genetic manipulation, and well-documented genome, *E. coli* is extensively studied [21,22]. *E. coli* exhibits a distinct cellular structure (Figure 1) comprising layers including the cytoplasm, plasma membrane (inner membrane), cell wall, and a unique outer membrane. The cytoplasm contains essential components for cellular functions and metabolism, while the plasma membrane acts as a selective barrier controlling molecular movement. The cell wall provides structural support and protection, while the outer membrane acts as an additional protective layer and regulates external substance entry [23].

Functionalizing CDs with biomolecules or antibodies enables the selective targeting of *E. coli*, enhancing detection specificity [24]. CDs also possess high photostability and brightness, ensuring robust fluorescence signals during imaging and detection assays. Their nano-meter size facilitates efficient cellular uptake and distribution, allowing for the effective labeling and tracking of *E. coli* within biological samples. Furthermore, CDs can be synthesized through straightforward and cost-effective methods, making them suitable for various applications [25].

Fluorescence spectroscopy and time-resolved fluorescence techniques have become powerful tools for bacterial characterization, surpassing traditional methods. They offer real-time analysis, label-free detection, and the ability to use specific fluorescent labels for targeted analysis [15]. These techniques are highly sensitive, enabling the detection of low bacterial concentrations and facilitating early infection detection. Fluorescent probes targeting specific bacterial components or biomarkers provide excellent specificity for distinguishing bacterial species [26]. In addition, fluorescence imaging allows for the visualization of fluorescent molecules, providing insights into cell function and interactions at the single-molecule level [27]. Extensive studies using fluorescence-based techniques have been conducted on *E. coli* [28], providing valuable insights into fundamental biological processes applicable to other organisms. Such investigations can lead to the development of novel detection strategies for different bacterial species [29,30].

Fluorescence anisotropy (FA) and FA decay have become valuable techniques for the detection and characterization of bacteria, including *E. coli* [30]. FA enables the measurement of polarization and offers insights into molecular rotation and fluorophore mobility. Meanwhile, FA decay provides valuable information about the presence of multiple rotational correlation times (*θ*’s), indicating heterogeneity within the molecular population and variations in the size, shape, and internal motions of the fluorophore–biomolecule conjugate [31]. CDs, due to their smaller size and unique properties, possess the ability to selectively bind to bacterial targets at different locations, as well as different biomolecules within the cell interior. By analyzing the changes in FA and *θ* at these locations, valuable information regarding alterations in local microviscosity or factors impeding mobility, such as aggregation or biomolecular interactions, can be obtained. This approach enhances our understanding and monitoring of bacterial behavior, revealing insights into intracellular interactions and dynamics [32].

Time-resolved fluorescence measurements can be performed in the time domain (TD) or the frequency domain (FD), both interconnected through the Fourier transform. While TD fluorometry is more common and offers higher-time resolution and more robustness to sample heterogeneity [33,34], FD offers high speed, simplicity, flexibility, robustness, and reduced phototoxicity for clinical applications. Despite drawbacks, FD fluorometers are routinely used to study complex FI and FA decays. Continuous improvements in FD fluorometers show promise for the technique’s future [33,35,36].

Previously, we demonstrated a fast method using FD fluorescence lifetime (FLT) imaging microscopy (FLIM) for detecting bacterial and viral pathogens, including *E. coli* [37,38]. FLT offers a distinctive approach to investigating complex biological environments and uncovering molecular heterogeneity by tracking differences in the excited state kinetics of one or more fluorophores [27]. Recently, we reported a FD TR-FAIM system to extract the maps of several fluorescence characteristics: FI, FLT, FA, and *θ* of two types of CDs. The link of CDs to gold nanoparticles was detected by increased FA and *θ* [39]. In this study, we examined the pH- and polarity-responsive properties of newly developed nanomaterials like CDs through FLIM analysis. In addition, we explored the changes in viscosity within the environment of CDs using TR-FAIM. Furthermore, we utilized FLIM to sense the intracellular pH and polarity of *E. coli* using CDs, while TR-FAIM was employed to study CD aggregations and biomolecular interactions within *E. coli* cells.

## 2. Materials and Methods

This section discusses the specific preparation procedures for both the CDs and bacterial samples. Furthermore, it covers the measurement procedure, calibration, and data analysis techniques employed to obtain experimental results.

### 2.1. Sample Preparation

#### 2.1.1. Preparation of CDs

Thermally synthesized solvent CDs were given an alkali treatment to produce the metal–cation-functionalized CDs. In further details, urea (2 g) and citric acid (1 g) were heated to 160 °C under solvothermal conditions for 6 h, and then the mixture was cooled to room temperature. The resulting dark brown solution was combined with 20 mL of an alkali (NaOH) aqueous solution (50 mg mL^−1^), agitated for 1 min, then centrifuged at 16,000 r min^−1^ for 10 min. The supernatant was removed and solution was left to evaporate with dimethylformamide (DMF) completely. For pH studies, citrate buffer (pH 4) and phosphate buffer (pH 7) were utilized as the buffer solutions. The minimum quantities of NaOH or HCl were added to the solution to adjust pH variations. For the polarity studies, eleven solutions were prepared with varying percentages (0–100%) of a dioxane–water mixture, thereby changing polarity. The dioxane volume was increased from 0 to 100 mL (0 mL, 10 mL, 20 mL, etc.), while the water volume was simultaneously decreased from 100 mL to 0 (100 mL, 90 mL, 80 mL, etc.). The same amount of CDs were dissolved in each solution.

#### 2.1.2. Bacterial Sample Preparation

Gram-negative *E. coli* ATCC 25922 were cultivated overnight at 37 °C with 180 rpm agitation in a lysogeny broth (LB). By measuring absorbance at 595 nm (OD^595^), the resulting bacterial concentration in LB broth was adjusted to 5 × 10^7^. The usual procedure was to combine 500 µL of bacterial cells (5 × 10^7^) in LB with 500 µL of CDs and incubate the mixture at 37 °C for 18 h while shaking it at 120 rpm.

#### 2.1.3. Preparation of Fluorescein Solution

In this study, a fluorescein solution (50 µM) was employed as a reference for the FD-FLIM system calibration. The solution was prepared by dissolving fluorescein (Sigma, St. Louis, MO, USA) in a phosphate-buffered saline (Biological industries, Kibbutz Beit Haemek 25115, Israel) solution with a pH value of 7.4.

### 2.2. Experimental Procedure

#### 2.2.1. Experimental Setup

The instrumental setup for FD TR-FAIM (Figure 2) is based on Lambert instruments’ existing FD-FLIM technology (LIFA, Groningen, The Netherlands) [40]. FA and FA decay imaging are achieved by adding fixed linear polarizers (one horizontal and one vertical) in the excitation path and a polarizing beam splitter (PBS) in the emission path (Thorlabs Inc., Newton, NJ, USA). The PBS separates the fluorescence emission into orthogonal polarizations focused on different regions of the CCD camera for simultaneous parallel and perpendicular image recording.

The excitation source is a multi-LED with modulated sinusoidal wavelengths (403 nm, 468 nm, or 537 nm) controlled by a signal generator (Prior OptiScan I, Rockland, MA, USA). In this study, six modulation frequencies between 20 and 70 MHz were used. The device placed at the LED output has three options for sample excitation: vertical polarizer-modulated light (for FA or FLT measurements), horizontal polarizer-modulated light (for G-factor calculation support, as elaborated on in Section 2.2.2), and unpolarized-modulation light (for calibration).

An Olympus IX-81 inverted microscope (OLYMPUS, Tokyo, Japan) with a 10X, NA = 0.4 objective is employed for sample focusing, and a dichroic mirror within the microscope filters the excitation to transmit only the fluorescence emission to the PBS. The PBS splits the polarized fluorescence emission into vertical and horizontal orientations, which are recorded by the image intensifier. In addition to parallel and perpendicular intensity images, parallel and perpendicular images of both the phase shift and the modulation depth are extracted. These frequency response data (FRD) are acquired using a gain-modulated detection device (image intensifier) operating at the same frequency as the fluorescence emission with various phase angles (homodyne phase-sensitive detection) [40]. FRD is calculated per pixel using FLIM software (LI-FLIM), controlling the signal generator, LED, image intensifier, and camera settings, and presenting data as 2D images with a spatial resolution of 1392 × 1040 pixels. The integrated system provides a temporal resolution of approximately 80 ps for both FLT and *θ*, which is consistent with the manufacturer’s specifications [40], and the uncertainty in the FA value of (*r*) is estimated to be approximately ±0.001.

A pinhole is used to control the width of the FI emission and prevent overlap between the polarized beams at the input of the PBS. The output of the image intensifier is connected to a 16-bit CCD camera (LI^2^CAM MD), and the field of view (14.4 mm × 10.8 mm) is divided between the two polarization components of the emission. The two polarized emission components, parallel (||) and perpendicular (⊥), depend on the fluorescent material’s decay constants and the modulation frequency (represented by the angular modulation frequency, *ω*). They are modulated at the same excitation frequency, but with a phase shift between them. The AC amplitude of the perpendicular component ($I_{⊥}^{AC}$) leads the AC amplitude of the parallel component ($I_{\left|\right|}^{AC}$) by an angle $\Delta \phi =\phi_{⊥}-\phi_{∥}$. In addition, the perpendicular component ($I_{⊥}^{AC}$) is reduced relative to the parallel component ($I_{∥}^{AC}$) by the amplitude ratio $\Lambda_{\omega }$, calculated as the product of the modulation depth ratios and the DC ratios of the two polarized beams.
(1)$$\Lambda_{\omega }=\frac{I_{\left|\right|}^{AC}}{I_{⊥}^{AC}}=\frac{m_{\left|\right|}}{m_{⊥}}\frac{I_{\left|\right|}^{DC}}{I_{⊥}^{DC}},$$
The modulated anisotropy, *r_ω_*, can be calculated using Equation (1) with *ω* set to zero.
(2)$$r_{\omega }=\frac{\Lambda_{\omega }-1}{\Lambda_{\omega }+2}.$$
The fundamental principles of FD analysis were previously discussed [41].

#### 2.2.2. Calibration

All measurements were conducted at a standard ambient temperature of 20 °C. The FD-FLIM and FD TR-FAIM systems have slightly different configurations and distinct calibration processes.

To resolve FLT images, the FRD of the FI decays were extracted separately. This was accomplished through a dedicated experiment in which the PBS was replaced with a linear polarizer set at a specific angle of 54.70 degrees from the vertical (referred to as the magic angle setting), effectively eliminating any influence of FA on the extracted FRD [42]. To mitigate the frequency dependency of the system, a reference sample of fluorescein dye (Section 2.1.3) with a known FLT value was used for calibration (details of this method can be found in Appendix A).

In addition, to ensure an accurate calculation of DC and AC ratios between $I_{\left|\right|}$ and $I_{⊥}$, as well as obtaining accurate *r* and *θ* maps, the potential electrical or optical distortions in the polarization components need to be compensated for, including differences in transmission efficiencies. In our experimental setup using PBS and filters, the compensation factor (G factor) is typically insignificant. However, it should be noted that the reflected beam in the PBS exhibits a leakage of approximately 5% from the other polarization component across the wavelength range of 420 to 680 nm. The derivation of compensation formulas and determination of the G factor (1.05 ± 0.01) are described in Appendix A. The determination of the 5% leakage was previously described elsewhere [39]. Additionally, excitation with unpolarized light resulted in no phase shift difference between the two polarized components, eliminating the need for corrections for instrumental effects on the phase shift.

#### 2.2.3. Data Analysis

The FRD for both the FI and FA decays were analyzed using MATLAB 2021b software (MathWorks, Natick, MA, USA). The FI and FLT images were subjected to masking, excluding pixels with FI values lower than 33% of the maximum pixel value. Similarly, the *r* and *θ* images were masked by removing pixels below a threshold of 33% of the maximum FI values in each orientation. This masking approach was employed to eliminate regions with ambiguous or unreliable data, and masked the *E. coli* autofluorescence intensity, which was approximately one order of magnitude lower than the FI of CDs. As a result, the autofluorescence signal did not affect the FLIM measurement of *E. coli* treated with CDs in the FD-FLIM. In addition, the FA images were calculated after performing pixel registration on the two polarized emission images. This registration involved finding the center of gravity for each polarized component. The center of gravity of each image could be used to define a circular or square region of interest, depending on the type of requested sample. A circular region of interest with a radius of 40 pixels was selected for the extraction of images and their statistical properties in the case of CDs.

Generally, the mathematical model used to describe the total FI, *I*(*t*), of a sample exhibiting multi-FLT can be described by a sum of exponentials, with each exponential term representing a different decay component [41]:(3)$$I(t)=\sum_{i}\alpha_{i}e^{-t/\tau_{i}}$$
Factor *α_i_* denotes the species fraction of component *i* $\left(\underset{i}{\sum }\alpha_{i}=1\right),$ with an FLT of *τ_i_*, and *t* is the time after excitation. The analysis of CD FLT typically favored the triple exponential fitting model over both the single exponential and double exponential decay models. However, not all samples, encompassing CDs in different solvents with a varying pH or polarity, exhibited a distinct triple exponential decay pattern. This can be attributed to the challenges associated with triple exponential fitting. Nevertheless, we found that using the weighted average FLT as a descriptive parameter for CD samples in different-polarity solvents, and the longest and most dominant FLT component as a descriptive parameter for CD samples in solvents of various pHs, yields satisfactory results. The presence of the longest and most dominant FLT component was consistent across both the double and triple exponential fittings.

The average FLT ($\tau_{A}$) was determined by calculating the weighted average of the number of FLT components (*N*) using:(4)$$\tau_{A}=\sum_{i=1}^{N}\alpha_{i}\cdot \tau_{i},$$
To validate the general FI trends, as well as the multi-exponential FI decay law, a TD system was employed. The TD system was a time-correlated, single-photon counting (TCSPC, Becker and Hickel, Berlin, Germany), with a capability to measure both the FI and FLT.

Similar to the FI decay (Equation (3)), the FA decay can also be described as the sum of the exponential:(5)$$r(t)=\sum_{j}r_{0j}\cdot e^{-t/\theta_{j}}=r_{0}\sum_{j}g_{j}\cdot e^{-t/\theta_{j}}$$

The term *r*_0*j*_ denotes the amplitude, *t* = 0 of each component *j* in the FA decay with a rotational correlation time *θ_j_*. The term *r*_0_ is the fundamental FA and *g_j_* is the fractional amplitude for each component *j* in the FA decay where $\left(\underset{j}{\sum }g_{j}=1\right)$ [41].

## 3. Results and Discussion

This section includes the analysis of the absorption and emission spectra of CDs, *E. coli*, and the CD–*E. coli* complex. Next, it provides details regarding the specific locations of interaction between CDs and *E. coli*. Furthermore, it addresses the sensitivity of CDs to various pH and polarity solutions using FI and FA decay analyses. Finally, imaging of the FI and FA decay of CDs and the CD–*E. coli* complex are presented.

### 3.1. Investigating the Interaction between CDs and E. coli Bacteria

The polar bacterium undergoes polarity changes throughout its lifespan. Its membrane comprises a negatively charged head, a positively charged tail, and a hydrophobic, non-polar lipid bilayer in the cell wall. This lipid bilayer provides an ideal binding site for CDs, which are attracted to hydrophobic environments and contribute to the reduction in polarity. The electrostatic force draws CDs to the outer cell membrane, facilitated by the hydrophobic nature of the membrane’s head region [43,44].

Zeta-potential measurements are employed to confirm polarity changes within bacterial cells. The zeta potential, or electrokinetic potential, reflects the potential difference between the electric double layer of mobile particles and the surrounding dispersant layer at the slipping plane [45]. The zeta-potential measurements of CDs and *E. coli* were found to be −8.3 mV and −15.7 mV, respectively. Upon formation of the *E. coli*–CD complex, the zeta potential decreased to −2.32 mV, indicating a decrease in polarity, as anticipated. The zeta potential of CDs plays a vital role in the interaction with bacteria. When the zeta potential is excessively negative, CDs are unable to enter. Conversely, if the zeta potential is slightly negative, it fails to effectively repel CDs, enabling them to penetrate bacterial cells [46].

Previous studies have shown that CDs can bind to various locations within *E. coli*. CDs can get trapped in the outer cell wall through electrostatic attraction [14,47]. In addition, CDs of varying sizes and surface properties can also be internalized by *E. coli*, binding to the cytoplasmic membrane through processes like endocytosis. This allows CDs to interact with the membrane inner surface [48].

Figure 3 presents the morphology of CDs, *E. coli* bacterial cells in the control group, and those treated with CDs using transmission electron microscopy (TEM, JEM-1400Flash, Peabody, MA, USA). The HR-TEM image of CDs (Figure 3a) reveals their oval shape and crystalline nature. In the control group, bacterial cells exhibit a complete internal structure characterized by a well-connected cell wall and cell membrane (Figure 3b). However, in the presence of CDs, bacterial cells exhibited slight deformation, with numerous CDs accumulating on the bacterial cell walls (Figure 3c). Additionally, some CDs were observed to enter the cell interior and attach to the inner membrane, while others gathered within the cytoplasm (Figure 3d).

The interaction between CDs and *E. coli* is primarily electrostatic, without any chemical bonding occurring between them. As a result, the composition and structure of CDs remain unchanged after their interaction with *E. coli*. This is corroborated by the TEM image (Figure 3c), which demonstrates the integrity of CDs even after combining with *E. coli*.

### 3.2. Absorption and Emission Spectra of CDs, E. coli, and the CD–E. coli Complex

The absorption spectra were recorded using a UV-1900 spectrophotometer (Shimadzu, Kyoto, Japan) and the emission spectra using Cary Eclipse spectrofluorometer (Varian, Palo Alto, CA, USA).

It has been noted that polarity changes cause both the FI and the absorption and emission peaks to alter. Decreased solvent polarity causes a blue shift in the emission spectrum, while higher solvent polarity results in a red shift [49]. At 390 nm and 467 nm, CDs had two UV–VIS absorption peaks (Figure 4a). The peaks at 390 nm and 467 nm, respectively, correspond to the π–π* and n–π* transitions. In the transition from π to π*, the maximum value moves to a longer wavelength. Only the excited state will interact with the polar solvent, stabilizing it. Because of this, absorption moves to a longer wavelength. With increasing solvent polarity, n–π* transition-related peaks are pushed to shorter wavelengths (blue shift). This results from greater lone pair solvation, which reduces the n orbital energy.

In addition, the *E. coli* autofluorescence exhibited a weaker intensity on order of magnitude relating to the CD–*E. coli* complex (Figure 4b).

### 3.3. Investigation of the Polarity and the pH Responsiveness of CDs

To evaluate the suitability of our CDs as pH and polarity probes, we conducted two sets of experiments. In the first set, we adjusted the pH of the CD solutions by adding NaOH or HCl, covering a range from 2 to 14. In the second set, we modulated the polarity by gradually increasing the percentage of dioxane in dioxane–water solutions.

Table A1 in Appendix B displays the typical triple exponential fit obtained from both TD and FD-FLIM systems. The molecular states, aromatic domain states, and carbon core states are the three types of emission centers in CDs [50]. Our findings indicate that the dominant component (~7.8 ns) of CDs exhibited the highest pH sensitivity. As these long FLT-excited states are directly associated with molecular emissive states, they are sensitive to pH. Nevertheless, the weighted average FLT showed the most distinct trend for polarity responsivity. Consequently, we selected these parameters for our investigations.

#### 3.3.1. pH Responsivity of CDs

First, the FI and FLT of CDs to different pH values ranging from 2 to 14 were examined using the FD-FLIM system. To investigate pH-dependent effects, citrate buffer (pH 4) and phosphate buffer (pH 7) were used as the buffer solutions. The pH variations in the solutions were adjusted by adding small amounts of NaOH or HCl as necessary.

The results demonstrated that the FI of CDs (red squares, Figure 5a) exhibited a gradual change across the pH range of 2–14. In the acidic pH range (2–8), the FI increased by 42%, whereas in the alkaline pH range (10–14), the FI decreased by 32%. The FI trend was confirmed using the TD system measurements.

Furthermore, we observed a linear increase in the FLT of the CD solutions as the pH increased within the range of 2 to 8 (blue triangles, Figure 5b). The FLT values ranged from 6.38 ± 0.05 ns to 8.03 ± 0.21 ns. Consequently, within this pH range, the pH level of the CDs in the solutions can be determined by measuring the FLT and utilizing the linear regression equation $FLT\;\left[\mathrm{ns}\right]=0.29\cdot \mathrm{pH}+5.80$. However, due to challenges in accurately fitting the exponential decay model for FI at pH values above 8, we excluded the corresponding data points from our analysis. It is worth noting that *E. coli* has the ability to thrive in an environment within a pH range of 4.5 to 9, making pH values above 9 less significant for FLT analysis of *E. coli* [51,52]. To ensure enzyme activity and stability of proteins and nucleic acids, *E. coli* maintains its cytoplasmic pH between 7.2 and 7.8 throughout this broad pH range [52].

The mechanism underlying the fluorescence of CDs remains a subject of debate. However, several reports suggest that the protonation or deprotonation of the amino functional groups on the surface of the CDs can vary in different pH environments, potentially leading to changes in fluorescence [53,54,55]. For example, the addition of excess OH- ions combine with Na^+^ to produce NaOH in highly alkaline conditions may lead to a decline in FI [56].

#### 3.3.2. The Polarity Responsivity of CDs

To examine the FI and FLT of CDs in solvents of varying polarities, we employed the FD-FLIM system. Modifying the polarity was achieved by increasing the concentrations of dioxane (0–100%) in a dioxane–water solution.

Zeta-potential measurements were conducted to validate the reduction in polarity (purple asterixis, Figure 6a). The results indicated a significant decrease in zeta potential, from −8.23 mV to −0.23 mV, as the dioxane concentration increased from 0% to 50%. In this range of dioxane concentrations, both FI (red squares, Figure 6b) and FLT (blue triangles, Figure 6c) measurements exhibited a clear increase. Interestingly, at a dioxane concentration of 60%, there was a change in the sign of the zeta potential, accompanied by a change in the trend of both FI and FLT measurements. Furthermore, as the dioxane concentration increased from 60% to 100%, the zeta potential decreased from 1.56 mV to 0.05 mV. Correspondingly, both FI and FLT measurements showed a clear decrease within this range. It is worth mentioning that the *E. coli*, CDs, and the CD–*E. coli* complex displayed a negative zeta potential. Thus, their values fall within the range of the increasing FI and FLT observed in the measurements. For example, the CD–*E. coli* complex exhibited a zeta potential of −2.32 mV, which corresponds to a dioxane concentration of 30%.

The influence of dioxane concentration/polarity on the FI and FLT can be explained by several theories. Firstly, in highly polar solvents, strong solvent–fluorophore interactions stabilize the excited state, leading to reduced FI and FLT through enhanced nonradiative relaxation pathways. Conversely, decreasing solvent polarity weakens these interactions, resulting in increased FI and FLT by reducing solvation effects and slowing down nonradiative relaxation processes [57]. Moreover, the concentration of dioxane has been shown to enhance the quantum yield, causing simultaneous increases in the FI and FLT [49,57], suggesting possible quenching effects. Secondly, changes in solvent polarity induce shifts in the emission spectrum, with an increasing dioxane concentration causing a blue shift [58]. Thirdly, polarity influences the rates of excited-state processes such as internal conversion or intersystem crossing, which compete with fluorescence emission and affect the FI and FLT. Lastly, polar solvents may contain quenchers that suppress fluorescence through energy or electron transfer processes. Decreasing polarity reduces the presence or activity of these quenchers, resulting in a higher FI [2,58].

The transition of the zeta potential from negative to positive can have an impact on the trends of the FI and FLT through changes in the quantum yield [59]. This shift in the zeta potential indicates a change in the surface charge of the particles or molecules involved. Such a charge alteration can influence the interactions between the fluorophore and its surrounding environment, affecting factors like solvation, aggregation, or molecular interactions that contribute to the FI. Therefore, the change in the zeta potential may influence the observed FI and FLT trends. In addition, it was previously shown that the quantum yield increases when the zeta potential decreases with a negative sign [59]. This may also explain the received trend in the FI and FLT. The FLT decrease can also result from an increase in the index of refraction [60]. In addition, the decrease in the FI from 60% to 100% may result from a larger blue shift in the emission spectrum, as well as a decrease in the fluorescence emission, as seen in previous works about several fluorophores [61,62].

Finally, since both the FI and FLT exhibited complex behavior throughout the region of increasing the dioxane concentration, another approach was considered. Polarity may significantly impact FA and *θ*. Typically, in highly polar solvents, due to interactions with polar solvent molecules, restricted rotational freedom of the fluorophore leads to a higher alignment and a hence higher FA and *θ*. Conversely, reduced solvent polarity allows for greater rotational freedom, resulting in lower alignment and lower FA [63]. Moreover, polarity influences molecular interactions, with stronger binding in polar environments leading to an increased FA and *θ*. Thus, steady-state and time-resolved FA analyses were also performed to investigate the effect of dioxane percentage on FA.

Surprisingly, as the dioxane percentage increased from 0 to 100%, a moderate but consistent increase in FA was observed (red squares, Figure 7a), contrary to the expected trend of a decreased FA with reduced solvent polarity. In the same manner, *θ* values increased with the reduced polarity (green circles, Figure 7b), *r*_0_ remained constant throughout the 11 samples and was 0.38 ± 0.02. Unfortunately, the FA decay law for 100% dioxane was determined with less accuracy.

Considering the conflicting observations, an alternative theory was considered. The addition of dioxane solute to the solvent disrupts interactions between water solvent molecules, increasing the solvent viscosity [64]. Dioxane, with its relatively high molecular weight, forms strong intermolecular interactions with the solvent, hindering solvent movement and flow, and leading to increased viscosity directly related to the concentration of dioxane. The observed trend of an increasing FA with higher dioxane percentages suggests that the increase in viscosity masks the effect of the decrease in polarity. For the last three samples with dioxane percentages ranging from 80 to 100, the polarity remains relatively constant (Figure 6a), while the viscosity increases significantly [64]. This leads to a dramatic increase in the FA and *θ* overall. Nevertheless, although the FI and FLT exhibit a complex trend for increasing dioxane concentrations which reduced polarity, the FA and *θ* presented a clear trend, but due to the increased viscosity and not directly the decreased polarity.

It is important to mention that the oval shape of CDs led to the determination of a multi-exponential fit for the FA decay. Fortunately, one component of *θ* (0.5 ± 0.11 ns) consistently appeared with the highest prevalence in both the double and triple exponential decay fits. This particular component was selected for analysis. An example demonstrating the single, double, and triple exponential fits for CDs with 0% dioxane can be found in Table A2 in Appendix B.

### 3.4. Fluorescence Imaging of the CDs and the CD–E. coli Complex

To investigate the impact of the *E. coli* interaction with CDs on their properties, we conducted imaging experiments to map the key fluorescence characteristics of the CDs. These maps included the FI and FLT, as well as *r* and *θ* values (Figure 8). Subsequently, we generated similar images for the CD–*E. coli* complex to examine any changes induced by the interaction (Figure 9). Similar to the pH investigation in Section 3.3.1, the dominant FLT component (~7.8 ns) was used in the CD FLT imaging. The typical triple exponential fit for the three samples is presented in Appendix B.

The longer component FLT of CDs was 7.80 ± 0.07 ns (Figure 8b). The FA was 0.062 ± 0.019 (Figure 8c). The rotational correlation time values of the CDs alone exhibited two distinct components: 10 ± 0.47 ns and 0.5 ± 0.11 ns (Figure 8d), with the latter being the more dominant component. The fundamental FA was 0.39 ± 0.02 (Appendix B).

Prior to analyzing the fluorescence of CDs in *E. coli* cells, we compared the autofluorescence of control *E. coli* with that of *E. coli* treated with CDs. *E. coli* exhibits distinct FLTs attributed to various natural fluorophores within its cells, providing insights into its physiological and biochemical properties. However, *E. coli* autofluorescence has limitations in fluorescence-based methods [65]. Its relatively weak autofluorescence restricts sensitivity and the signal-to-noise ratio, necessitating higher excitation power and longer acquisition times, which can increase phototoxicity and photobleaching. Additionally, the broad emission spectrum of *E. coli* autofluorescence may overlap with fluorophores or CDs, complicating signal separation and analysis [27,66]. Moreover, due to the weak FI of these natural fluorophores and the presence of multiple FLTs, FLT analysis relying on *E. coli* autofluorescence may lack sufficient accuracy in probing changes in these multiple FLTs. As expected, the autofluorescence intensity of the cells was significantly weaker, being approximately one order of magnitude lower than the FI of CDs. As a result, unlike the TCSPC system, the FD-FLIM system could not accurately determine the multi-exponential decay law of the control *E. coli*. To ensure an accurate analysis of the CD–*E. coli* complex and eliminate the impact of *E. coli* autofluorescence intensity and FLT, a threshold of 33% of the maximal pixel value (approximately three times the autofluorescence intensity) was implemented. This threshold effectively masked the weak autofluorescence, allowing for a precise FLT characterization of the CD–*E. coli* complex in the FD-FLIM. The extracted FLT components of *E. coli* by the TCSPC system are presented in Appendix B.

The analysis of FI, FLT, *r*, and *θ* maps of the *E. coli* cell (Figure 9a–d) revealed a distinct structural pattern. The *E. coli* structure exhibits three main circles: the outer circle corresponds to the cell wall, the middle circle represents the plasma membrane, and the innermost circle encompasses the cytoplasm. These three locations were previously identified as the binding sites for the CDs using TEM (Figure 3).

The higher FI observed in the interior of the cell (Figure 9a) suggests that a majority of the CDs have entered the cell, as well as the aggregation of some of the CDs, as confirmed by the TEM image (Figure 3d). However, some CDs are bound to the cell wall and plasma membrane. In contrast, the FLT exhibits an opposite trend, being lower in the cytoplasm compared to the cell exterior (Figure 9b). Remarkably, the linear formula relating FLT to the pH (Section 3.3.1) indicates that the cytoplasmic FLT (8.02 ± 0.14 ns) corresponds to a pH of approximately 7.67 ± 0.48, consistent with the reported internal pH range of *E. coli* (7.2 to 7.9) [67]. This demonstrates the suitability of FLT as a pH sensor for *E. coli*, with the capability to monitor and determine the pH values. Conversely, the significantly higher FLT in the cell exterior regions cannot be attributed to a higher pH, but rather to a decrease in polarity. This is because typically, the periplasmic space situated between the inner and outer membranes of Gram-negative bacteria such as *E. coli* has a slightly acidic pH attributed to the presence of enzymes and transporters involved in nutrient uptake and cell wall synthesis [68]. This acidic pH within the periplasmic space contributes to the pH gradient observed throughout *E. coli* cells, with the cytoplasm maintaining a near-neutral pH [67]. Therefore, the observed increase in FLT in the periplasmic space may not be attributed to a higher pH level.

The distribution of charge or other molecular characteristics within *E. coli* cells and their surroundings is referred to as their polarity. Proteins, nucleic acids, and lipids are examples of polar substances found in *E. coli* cells that help maintain the overall polarity of the cell [69]. Cellular polarity is affected by the interactions between these molecules, which have different charge distributions, and the environment [43]. There may be less interaction between the CDs and surrounding polar molecules at the outer peripheral region of the bacterial cell compared to the core. Due to lesser quenching or energy transfer processes brought on by polar molecules, this decreased contact can lead to a longer FLT. On the other hand, the presence of water molecules and polar molecules inside the cell may result in a higher polarity there, leading to a shorter FLT related to the exterior of the cell [43].

The FA and correlation time imaging (Figure 9c,d) showed a consistent pattern with the FI, indicating higher values in the cytoplasm compared to the plasma membrane and cell wall. This can be explained by two main factors. Firstly, the TEM image (Figure 3) demonstrated that CDs entering the cell tend to aggregate, resulting in larger rotating units and contributing to an increased FA and correlation time. Secondly, the presence of abundant biomolecules such as proteins, nucleic acids, and lipids in the cell interior enables binding interactions with CDs, limiting their molecular rotation and leading to higher FA and correlation time values [70,71].

The correlation time exhibited a similar trend to the FA, providing detailed insights into the variations within the cell environment. Through this analysis, specific regions of interaction between CDs and the cell interior, as well as CD aggregations, were identified. In contrast to the rotational correlation times found in the CDs alone, the CD–*E. coli* complex showed the same 10 ns component, but the dominant and shorter component increased dramatically from 0.5 ns in the CDs alone to 2.04 ± 1.06 ns in the CD–*E. coli* complex, reflecting its large heterogeneity.

In addition, the distribution of CDs within *E. coli* can be attributed to variations in density among different bacterial regions. Initially, it was suspected that the higher FA and correlation time observed in the cytoplasm could be attributed to its higher viscosity compared to the cell wall and plasma membrane. However, it is important to note that the density of live *E. coli* bacteria was reported to be 1.1 to 1.15 g/cm^3^ [72], whereas the density of the bacterial membrane/cell wall is measured at 1.302 g/cm^3^ [73]. Upon entry into the *E. coli* cell, CDs primarily accumulate near the denser bacterial DNA, which has a density of approximately 1.7 g/cm^3^ [74]. This suggests that the increase in FA and correlation time inside *E. coli* is not necessarily due to viscosity differences between the cytoplasm and the cell wall, but rather the result of biomolecular interactions and CD aggregations inside the cell.

Finally, multiple theories can explain the polarity changes observed in *E. coli*. Firstly, different bacterial structures, such as the cell wall and membrane, exhibit varying polarities. The cell membrane, with its polar phospholipid heads, can be considered more polar. Moreover, within the bacterial cytoplasm, there are distinct poles facilitated by the bacterial Min system [75]. The MinC and MinD proteins interact to form a complex that exhibits periodic oscillations between the poles approximately every 20 s [44]. Lastly, particle aggregates are found in condensed areas (such as bacterial DNA) of the cell, as shown in the inset of Figure 3b.

## 4. Conclusions

Recent advancements in time-resolved fluorescence imaging systems for clinical applications have shown promising results, although their availability remains limited. Our research group has collaborated with medical doctors and researchers from various disciplines to demonstrate the potential of FLIM in clinical settings, including identifying chromosomal abnormalities [76], detecting metastatic cells [77], BRCA2 mutation [78], and diagnosing thyroid tissues [79]. FA decay is another well-established technique with advantages such as high sensitivity, rapid analysis, and versatility in sample types [28,33,80,81]. These features make it a promising diagnostic tool for accurate and quick clinical diagnosis. While the widespread adoption of these systems in clinical practice may still be in the future, the considerable progress and potential for further advancements make them highly promising for medical diagnostics.

In this study, we employed innovative nanomaterial such as CDs that exhibit remarkable sensitivity to the pH, polarity, and viscosity of their surroundings. This sensitivity was investigated through FI decay and FA decay analyses, enabling us to gain insights into the unique properties of CDs and their interactions with *E. coli* at different locations.

The nanoscale size and distinctive characteristics of CDs allowed for their specific binding to *E. coli* at three distinct sites: the cytoplasm, the inner membrane, and the cell wall. This binding capability facilitated the visualization of *E. coli*’s structural features and provided an opportunity to explore environmental variations across these locations using FI and FA decay imaging techniques. This innovative approach revealed novel insights into the behavior of CDs within the *E. coli* environment.

Our investigation revealed complex relationships between the FI and the pH of CD surroundings. However, a linear relationship was established between the FLT and the pH range of 2–8 in the CD environment. Interestingly, both the FI and FLT analyses exhibited intricate relations, with decreasing polarity induced by an increasing dioxane concentration. To further understand these dynamics, FA decay investigations were performed, which showed unexpected results. While it is commonly understood that reduced polarity leads to a decrease in FA and rotational correlation time, our observations indicated an increase in both FA and rotational correlation time with increasing dioxane concentrations. This unexpected finding was attributed to the simultaneous increase in viscosity as the dioxane concentrations rose.

Subsequently, we applied FI and FA decay measurements to analyze *E. coli*. Notably, higher FI, FA, and correlation time values were observed in the cytoplasm compared to the peripheral areas, indicating a higher concentration of CDs and their successful penetration into the cell. The increased FA and rotational correlation time suggested the presence of biomolecular interactions, hindering the rotation of CDs and promoting their aggregation. The use of correlation time imaging allowed for the detailed visualization of these interactions and highlighted their specific locations.

By establishing a linear relationship between the FLT and pH in the CD environment, we successfully determined the pH of the *E. coli* cytoplasm, which aligned with the pH values reported in the literature. Furthermore, the higher FLT values observed in the periplasmic space were attributed to a decrease in polarity rather than an increase in pH, as the periplasmic space typically exhibits a slightly more acidic environment.

The combination of CDs with *E. coli* has promising applications in identifying biological processes involving pH and polarity changes. By using fluorescence imaging techniques like FI and FA decay imaging, comprehensive characterization of *E. coli* structures and their interactions with CDs has been achieved. These findings highlight the potential of fluorescence-based technologies for detecting and characterizing *E. coli*, as well as for applications in biosensing, bioimaging, antibacterial activities, and related fields. Furthermore, incorporating FLT and FA measurements enhances the capabilities of biological logic gates based on CDs.

## Figures and Tables

**Figure 1 nanomaterials-13-02068-f001:**
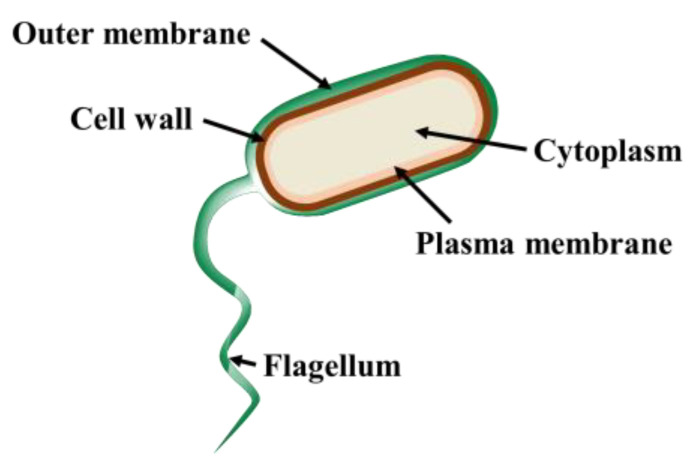
The structure of *E. coli* can be schematically depicted as consisting of distinct layers. These layers include the cytoplasm, plasma membrane, cell wall, and outer membrane, each serving specific functions within the bacterium’s cellular structure.

**Figure 2 nanomaterials-13-02068-f002:**
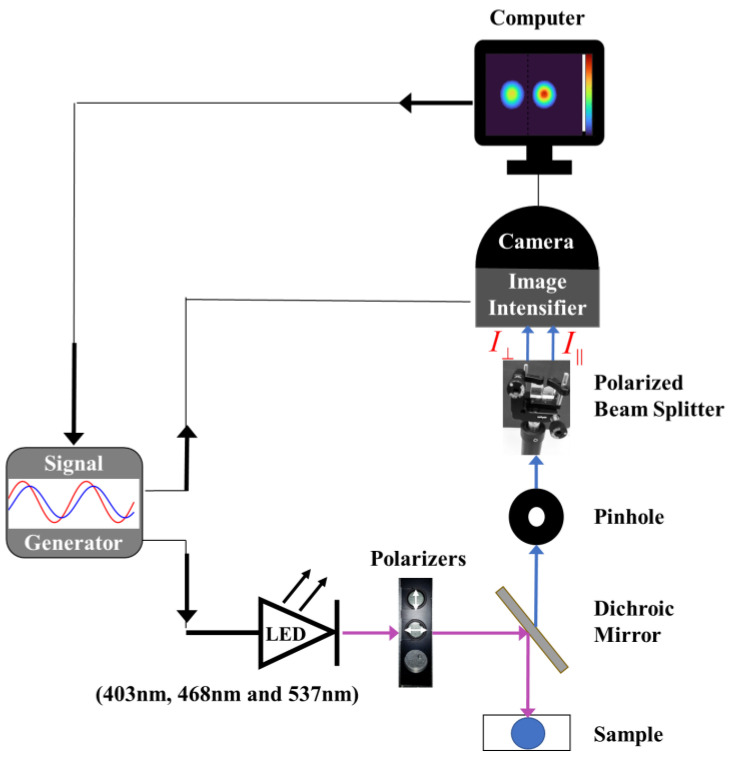
The FD TR-FAIM system is illustrated schematically. FA measurements are implemented by adding entrance polarizers (horizontal and vertical) at the output of the multi-LED source, along with a PBS at the input of the image intensifier in our FD-FLIM system. A pinhole at the PBS input adjusts the fluorescence image width to prevent overlap between the two polarization components. By using a mirror, the two polarized beams extracted by the PBS are directed parallel to the CCD camera, which divides its field of view between the two beams.

**Figure 3 nanomaterials-13-02068-f003:**
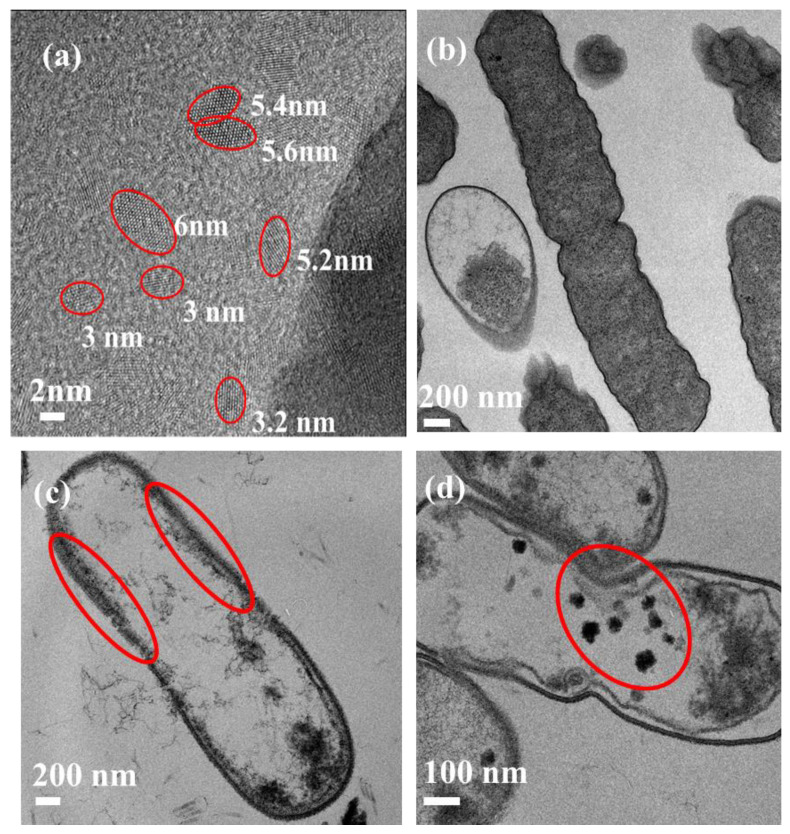
TEM image of CDs; (**a**) *E. coli*; (**b**) CDs bound to *E. coli*; (**c**) aggregates of CDs inside *E. coli* (**d**). The red circles mark CDs and their aggregates.

**Figure 4 nanomaterials-13-02068-f004:**
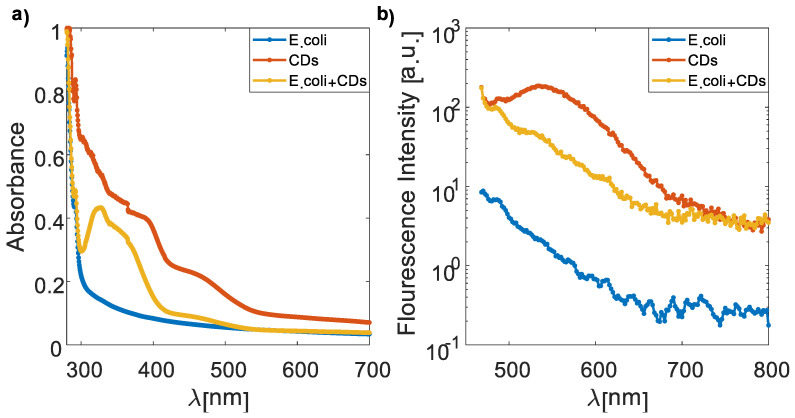
UV–VIS absorption spectra (**a**) and emission spectra (**b**) of the *E. coli*, CDs, and the CD–*E. coli* complex. Clearly, the *E. coli* autofluorescence intensity is about 1–2 orders of magnitude lower than the FI of CDs and the CD–*E. coli* complex.

**Figure 5 nanomaterials-13-02068-f005:**
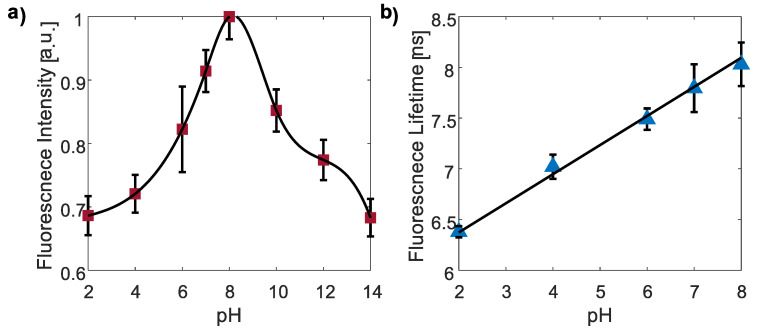
The FI (**a**) and FLT (**b**) of CDs at various pH values using the FD-FLIM system. At an acidic pH (2–8), the FI increased, while at an alkaline pH (10–14), the FI decreased (red squares). The FLT gradually increased linearly with the increasing pH values, while it showed a potential decrease when the pH exceeded 8.0 (blue triangles). Due to the difficulty in determining the exponential fitting for the FI decay model at pH values above 8, the corresponding data points were omitted from the analysis.

**Figure 6 nanomaterials-13-02068-f006:**
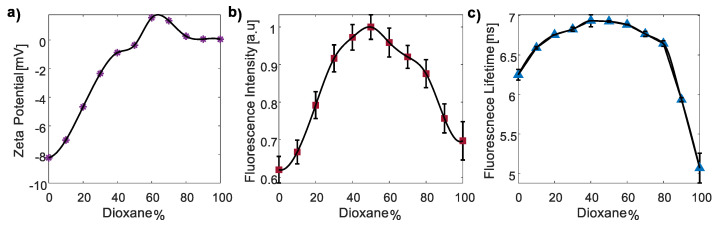
The zeta potential (**a**), FI measurements (**b**), and FLT measurements. (**c**) Measurements of the CDs in water–dioxane mixtures with increasing dioxane concentrations, from 0% to 100%. In a dioxane concentration of up to 50%, both the FI and apparent FLT increased, while beyond 50%, they both decreased.

**Figure 7 nanomaterials-13-02068-f007:**
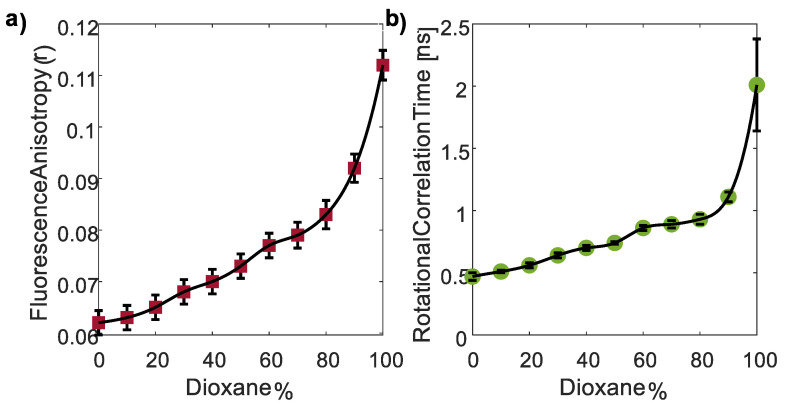
FA (**a**) and *θ* (**b**) measurements of CDs in water–dioxane mixtures with increasing dioxane concentrations. Surprisingly, the FA (*r*) and *θ* values increased as solvent polarity decreased. This observation suggests the involvement of an additional factor contributing to this trend, potentially viscosity.

**Figure 8 nanomaterials-13-02068-f008:**
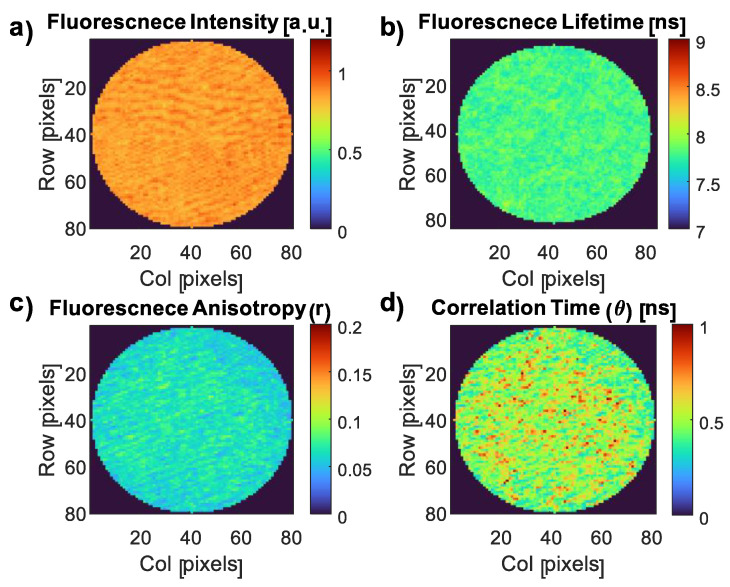
FI (**a**), FLT (**b**), *r* (**c**), and *θ* (**d**) images of CDs.

**Figure 9 nanomaterials-13-02068-f009:**
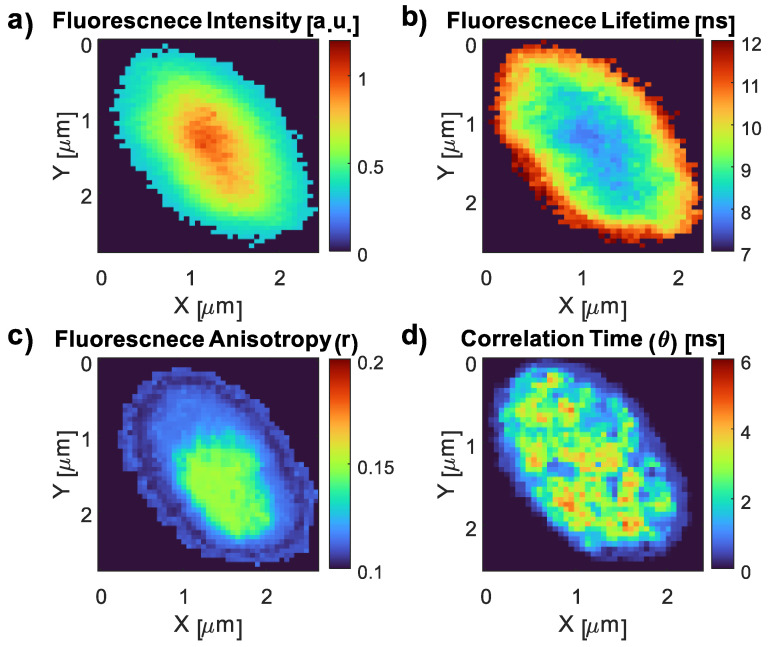
FI (**a**), FLT (**b**), *r* (**c**), and *θ* (**d**) images of the CD–*E. coli* complex.

## Data Availability

The data supporting this article are available in Figure 3, Figure 4, Figure 5, Figure 6, Figure 7, Figure 8 and Figure 9 and Table A1 and Table A2. The data sets analyzed in the present study are available from the corresponding author upon request.

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
