# Peer review of "Probing Polarity and pH Sensitivity of Carbon Dots in *Escherichia coli* through Time-Resolved Fluorescence Analyses"

_nanomaterials, 2023, doi:10.3390/nano13142068_

Round 1
Reviewer 1 Report
This submission described the development of a Carbon Dots-E. coli combination for probing the polarity and pH sensitivity in E. coli. It is an interesting work and the experiments were organized well. I am interested in several specific minor points.
1. The application of this system. Maybe the author should clarify the application of this system. Whether it was designed for the inner sensitivity of E. coli or the surrounding environment.
2. The consistency of this probing system. Since E. coli was selected to reflect the fluorescence signals, the consistency might be explained. For the inner sensitivity of E. coli, the cytoplasm and membranes showed different signals. For the surrounding, different size and component of the bacteria might affect the signals. Would the authors average the values or output the patterns with different signals?
3. I recommend the addition of the distribution of composition after the preparation of the Carbon Dots-E. coli combination. Further understanding of the characteristics of the system might enhance the potential of this technique.
Minor editing is needed.
Author Response
Reviewer 1
The application of this system. Maybe the author should clarify the application of this system. Whether it was designed for the inner sensitivity of E. coli or the surrounding environment.
We express our sincere gratitude to the reviewer for providing valuable comments on our manuscript. As a demonstration of its capabilities, we specifically chose E. coli bacteria as a model organism to highlight the effectiveness of the method described in our study. The unique properties of carbon dots make them highly promising for precise monitoring of changes in pH and polarity in biological environments. In addition, the selected system of CDs-E. coli demonstrates promising potential for various applications, including biosensing, bioimaging, and antibacterial activities. Moreover, there is a considerable research focus on utilizing carbon dots for the development of biological logic gates. By integrating information on fluorescence lifetime and fluorescence anisotropy, the capabilities and potential applications of these biological logic gates can be significantly enhanced.
In response to the valuable comments provided by the reviewer, we have made revisions to the manuscript to address his concerns. Specifically, we have incorporated a clarification in the manuscript to elucidate the main potential applications of the CDs-E. coli systems (marked in red in page 2 lines 54-58 in the revised manuscript). In addition, we have added a concluding paragraph to the end of the conclusion section, highlighting the main potential applications that can be derived from our study (marked in red in page 16, lines 629-636). These modifications aim to provide a clearer understanding of the significance and potential impact of our research findings.
The consistency of this probing system. Since E. coli was selected to reflect the fluorescence signals, the consistency might be explained. For the inner sensitivity of E. coli, the cytoplasm and membranes showed different signals. For the surrounding, different size and component of the bacteria might affect the signals. Would the authors average the values or output the patterns with different signals?
We express our gratitude to the reviewer for bringing this issue to our attention. We would like to clarify that the size did not have a significant impact on the results obtained in our study. Averaging the results for bacteria detection in different locations poses no issues, and we successfully distinguished distinct areas within the bacteria. However, it is crucial to highlight that the high spatial resolution was vital, particularly in classifying the structure of E. coli into different parts. It is important to note that the results presented in the manuscript regarding pH sensing in the cytoplasm were indeed obtained as an average, ensuring separate characterization of the cytoplasm alone (page 13, line 526 in the revised manuscript).
I recommend the addition of the distribution of composition after the preparation of the Carbon Dots-E. coli combination. Further understanding of the characteristics of the system might enhance the potential of this technique.
We express our gratitude to the reviewer for providing valuable comments. The composition of carbon dots remains unchanged after their interaction with E. coli. The interaction between the carbon dots and E. coli is primarily electrostatic, meaning that no chemical bonding occurs between them. As a result, the structure and composition of the carbon dots remain unmodified. This is supported by the TEM image, which confirms that the carbon dots remain intact even after combining with E. coli.
(a) TEM images of CDs bound to E. coli (b) Magnified image of the E. coli showing that the structure of CDs is intact after entering into the bacteria.
We would also like to provide further clarification regarding the "Carbon Dots-E. coli combination" mentioned in our study. It is important to note that there was no specific preparation conducted for this combination. The term merely describes the presence of carbon dots that were either attached to or penetrated the bacterial cell through natural chemical and physical processes.
Based on the reviewer suggestion we added a paragraph that elaborate on this issue (marked in red, page 8 lines 319-323)

Reviewer 2 Report
This study employed carbon dots (CDs) as pH- and polarity-sensitive nanomaterials for intracellular sensing. We utilized integrated time-resolved fluorescence anisotropy imaging (TR-FAIM) and fluorescence lifetime imaging microscopy (FLIM) techniques to comprehensively characterize the CDs' behavior.
How did the fluorescence intensity and fluorescence lifetime of carbon dots (CDs) change with pH and polarity?
What changes were observed in fluorescence anisotropy and rotational correlation time with decreasing polarity?
What future research and applications can be explored based on the findings of this study?
The work, in my opinion, is well-substantiated and carefully carried out. Interesting new results have been obtained. Therefore, I recommend this manuscript for publication.
Author Response
Reviewer 2
1. How did the fluorescence intensity and fluorescence lifetime of carbon dots (CDs) change with pH and polarity?
We thank the reviewer for the question. The fluorescence intensity (FI) and fluorescence lifetime (FLT) showed a distinct increase from pH 2 to pH 8. These pH values align well with the reported intracellular pH range of E. coli, which is typically in a range of 7.2 to 7.9. However, the FI exhibited a noticeable decrease beyond pH 8, while the multi-exponential fit for the FLT did not yield clear results. Additionally, the polarity analysis demonstrated that decreasing polarity within the negative zeta potential range resulted in an increase in both FI and FLT. Conversely, when the zeta potential changed to positive, both FI and FLT decreased as the polarity decreased (please refer to figure 5 and figure 6 in the revised manuscript).
What changes were observed in fluorescence anisotropy and rotational correlation time with decreasing polarity?
We thank the reviewer for the question. In polar solvents, the fluorophore experiences restricted rotational freedom due to interactions with solvent molecules, leading to higher fluorescence anisotropy (FA) and rotational correlation time (θ). Conversely, in less polar solvents, the fluorophore has greater rotational freedom, resulting in lower FA and θ. Surprisingly, when the dioxane percentage was increased from 0 to 100% and hence the polarity was decreased, a moderate but consistent increase in both FA and θ was observed (please refer to figure 7 in the revised manuscript), contrary to the expected trend of decreased FA and θ with reduced solvent polarity. To explain these conflicting observations, an alternative theory was considered. The addition of dioxane disrupts the interactions between water solvent molecules, leading to an increase in solvent viscosity. The strong intermolecular interactions between dioxane and the solvent hinder solvent movement and flow, resulting in increased viscosity that is directly related to the dioxane concentration. The observed trend of increasing FA and θ with higher dioxane percentages suggests that the increase in viscosity masks the effect of the decrease in polarity.
What future research and applications can be explored based on the findings of this study?
We thank the reviewer for pointing us into this lack of clarity. The potential applications of carbon dots are vast, offering numerous possibilities. Their high biocompatibility and low toxicity make them suitable for identifying various biological processes that involve changes in pH and polarity, such as those occurring in bacteria, biofilms, tumors, and other biological systems. The chosen CDs-E. coli system holds promising potential for applications as a biosensor, bioimaging tool, and antibacterial agent. Furthermore, there is significant research interest in developing biological logic gates using carbon dots. Incorporating fluorescence lifetime and fluorescence anisotropy information besides the conventional fluorescence intensity can enhance the capabilities and potential applications of these biological logic gates. Based on the reviewer comments, we have added a concluding paragraph to the end of the conclusion section, highlighting the main potential applications that can be derived from our study (marked in red in page 16, lines 629-636). This modification aims to provide a clearer understanding of the significance and potential impact of our research findings.